# An intrauterine cavity morcellator: A novel approach to high volume uterus morcellation. Ex-vivo study

Meir Pomeranz[1,2], Ron Schonman[1,2], Yael Yagur[1,2], Rina Tamir Yaniv[1,2], Zvi Klein[1,2], Yair Daykan[1,2,3]*

1 Department of Obstetrics and Gynecology, Meir Medical Center, Kfar Saba, Israel, 2 Sackler School of Medicine, Tel Aviv University, Tel Aviv, Israel, 3 Department of Urogynaecology, Cork University Maternity Hospital, Cork, Ireland

* yair.dykan@gmail.com

**Data Availability Statement:** All relevant data are within the paper and its Supporting Information files.

## Abstract

### Objective

Uterine size is one of the essential factors determining the feasibility of a minimally invasive gynecologic surgery approach. A traditional electromechanical morcellator is a well-known tool but not without flaws. We aim to assess feasibility and safety of a novel intrauterine power morcellation device for uterine size reduction to overcome these limitations during hysterectomy.

### Methods

This single-arm, observational study was conducted in a single tertiary care medical center from April 2022 to July 2022. Feasibility and safety of a novel intrauterine morcellation device for uterine size reduction was tested in ten post-hysterectomy uteri (Ex-vivo).

### Measurements and main results

Ten uteri were examined in this trial. No major complications occurred during the procedure. All ten (10) uteri were successfully reduced in size (size reduction range was between 9% to 54%). The average resection time using the Heracure Device was 4.3 minutes (range: 1min– 10min). Mean uterus weight reduction was 21%, with a mean circumference reduction of 25%. No leakage was observed from the outer surface of the uterus/serosa after the saline injection post-procedure examination.

### Conclusion

In this novel experiment, we verified the feasibility and safety of the Heracure device for vaginal intra-uterine morcellation for uterine size reduction. This technique could enable rapid and easy removal of the uterus through the vaginal orifice.

### Clinical trial registration

Name of the registry: ClinicalTrials.gov; Number Identifier: NCT05332132.

**Funding:** Yes The institute (hospital) received payment fees/devices from Heracure Medical Ltd. to support patient participation in the study https://www.linkedin.com/company/heracure-medical-ltd The authors did not receive personal payment. The founder has no role in study design, data collection and analysis, decision to publish, or preparation of the manuscript.

**Competing interests:** The authors have declared that no competing interests exist.

## Introduction

Minimally invasive gynecologic surgery is a major surgical technique that offers advantages over laparotomy, such as minimized use of narcotics, less intraoperative blood loss, shorter operative time, and rapid recovery [1–4]. Vaginal hysterectomy is the preferred approach for most patients [5]. Uterine size is one of the main parameters in determining the hysterectomy surgical approach: a uterine measurement of $\sim \leq$ 16 weeks and some degree of prolapse allows, in most cases, for a vaginal approach [6]. Laparoscopic approach is employed for cases of non-prolapsed uterus and cases with uterine measurement of less than 18–20 weeks [1, 5].

In some cases of enlarged uterine mass, morcellation is needed [7–9] and performed with the help of the in-bag (contained) morcellation [10]. For ~100 years manual morcellation was performed using a scalpel; while modern morcellation uses electromechanical morcellators that rapidly remove specimens, through the small laparoscopic incision, thus benefiting from the advantages of minimally invasive gynecologic surgery [11]. However, use of power morcellation in the peritoneal cavity was found to increase the risk of both benign and malignant cell dispersion [12, 13]. In 2020 the USA's Food and Drug Administration (FDA) released a safety update warning [14] limiting the use of laparoscopic power morcellation to certain appropriately selected women undergoing myomectomy or hysterectomy; when morcellation is appropriate, only contained morcellation should be performed. This warning aims to reduce the risk of disseminating malignant tissue in the treatment of benign-looking uterine fibroids. The immediate effect was a decline in the overall rate of minimally invasive surgery for these indications, as well as a sharp decline in the use of intraperitoneal morcellation [15, 16].

To maintain use of minimally invasive laparoscopic procedures solutions for in-bag power morcellation have evolved, beyond the use of scalpel morcellation [14]. Nonetheless, some limitations inherent to the use of in-bag power morcellation were shown in previous studies: this technique is cumbersome, time-consuming, with risk of perforation of the bag [17–19].

Therefore, we were eager to find a solution for the power morcellation technique suitable for minimally invasive laparoscopic or vaginal surgeries that will provide the patient with the best surgical technique, with the lowest risk rate and will maximize her recovery rate. The aim of this study was to evaluate the safety and feasibility of the intra-uterine morcellation device for uterine size reduction, to overcome these limitations during laparoscopic/vaginal hysterectomy.

## Materials and methods

This observational study was conducted in a single tertiary care medical center from April 18, 2022, to July 31, 2022. Participants were recruited from a list of patients consented for hysterectomy. Feasibility assessment of the morcellation device was employed on the post hysterectomy uterus (Ex-vivo) to reduce the uterine size. No intervention was applied in the patients, as the study only investigated the post-hysterectomy uterus.

### Study endpoints

The primary outcome evaluated was the procedure's safety. The main adverse effect was defined as uterine perforation at the end of procedure, which was assessed by inflating the uterus with saline.

Secondary outcomes were the device's success, which was defined as the ability to reduce uterus size/circumference and the time required for performing the morcellation procedure.

### Ethics approval and informed consent

The study was approved by the Meir Medical Center Ethics Committee in July 2021; approval number MMC- 0335–20 and in the ClinicalTrials.gov (Identifier: NCT05332132). Each patient

signed and dated a written informed consent form (ICF) prior to the procedure. The authors confirm that all ongoing and related trials for this intervention are registered.

Study protocol is attached separately.

This study used the implementation of enhancing the Quality and Transparency of health Research (EQUATOR) network guidelines.

## Case selection

The author performed 10 hysterectomies and performed morcellation on their uterus post-surgery, to evaluate the feasibility of this novel technique (Ex-vivo experiment). Flow diagram of the study is shown in Fig 1. Patients were selected based on the following inclusion criteria: undergoing hysterectomy (vaginal, laparoscopic, abdominal) due to benign gynecologic disease; Exclusion criteria: suspicion for malignancy (on ultrasound, prior endometrial biopsy, MRI).

The following patient and peri-procedure data were collected and retrospectively analyzed: Uterus size and volume, the duration of morcellation for uterine size reduction and perforation rate. The duration of the morcellation was defined as the time from the placement of the morcellator to the end of the procedure (reduction of uterus size). Uterus perforation by the Heracure device was an intra-procedure complication.

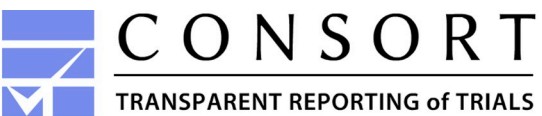

**CONSORT 2010 Flow Diagram**

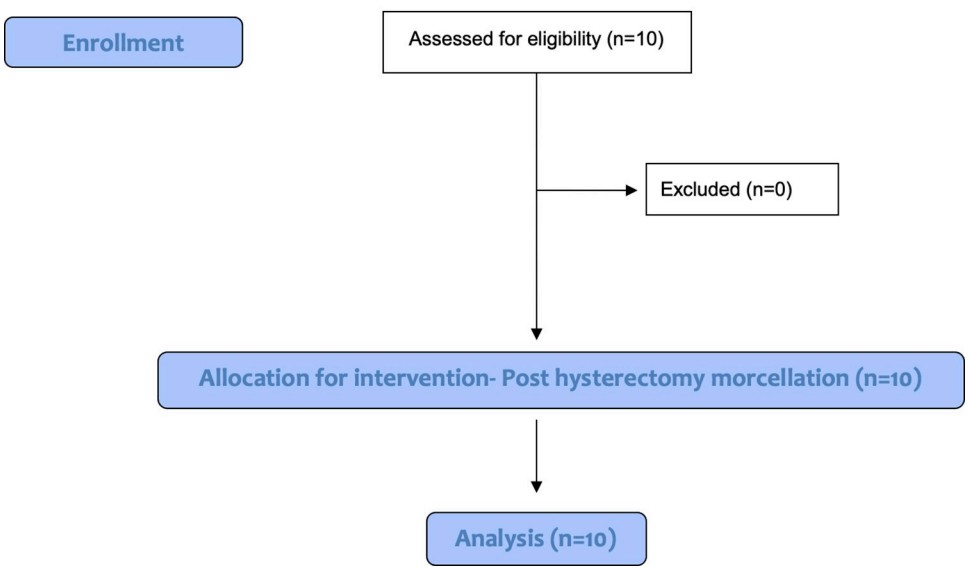

**Fig 1. CONSORT flow diagram.** (CONSORT 2010 flow diagram).

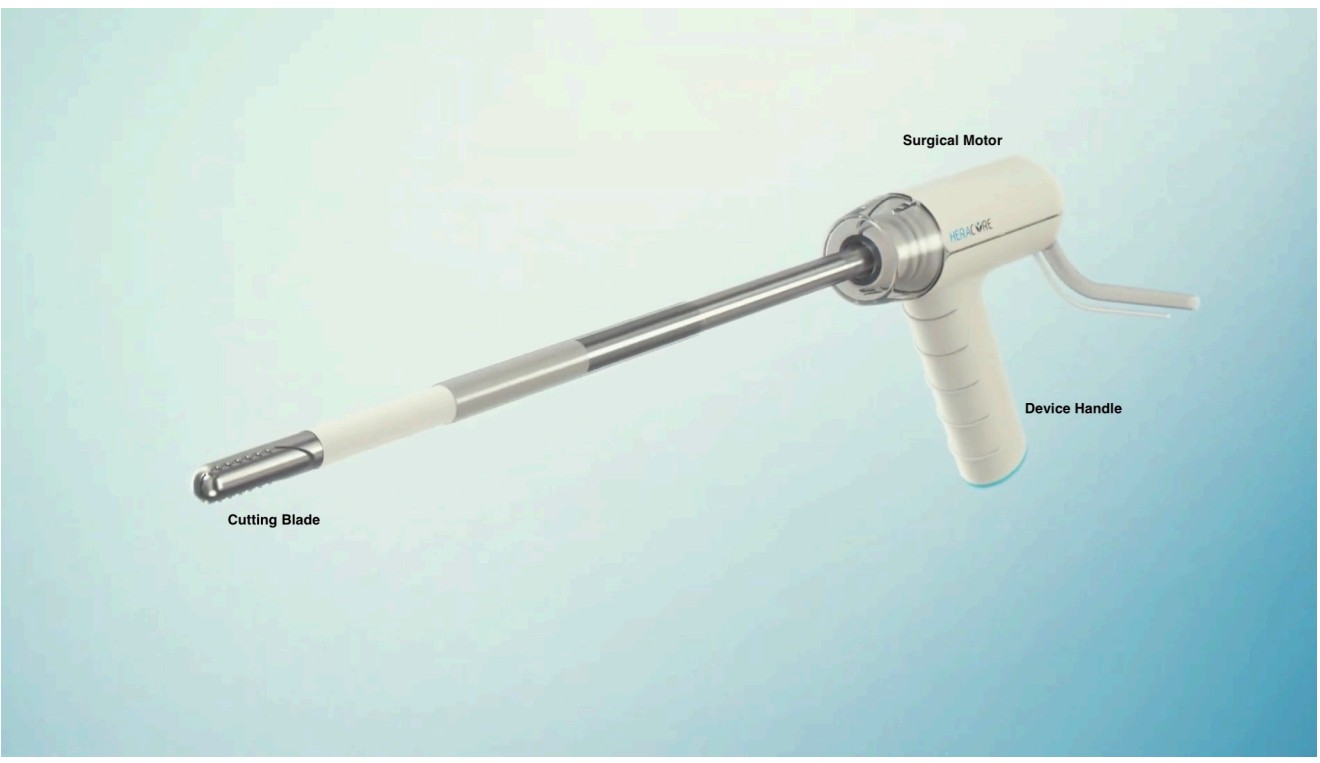

**Fig 2. The Heracure resection device.** (Fig 2, Republished from Heracure CP-0300 under a CC BY license, with permission from Heracure company, original copyright 2023).

## Morcellation device description

The Heracure System consists of the following components:

- Heracure Resection Device (Fig 2)

- Off the shelf FDA cleared motor control unit and foot pedal

- Off the shelf FDA cleared external vacuum source device with sterile tubing sets

The Control Unit contains FDA cleared electric motor and firmware motor controller that drives the Heracure Resection Device. The Control Unit is activated and deactivated by a foot pedal. The resection device features a rotating/oscillating side window cutter. The resected tissue particles and the irrigation fluid are transported using a vacuum source to a distal tissue collection container.

## Morcellation procedural technique

The study was conducted on 10 extirpated uteri after the end of each hysterectomy (Ex-vivo). The weight and size of each extirpated uterus was recorded prior to procedure.

Prior to Heracure device deployment, the cervix was dilated using Hagar dilators of max 13mm. The Heracure resection device was inserted into the uterine cavity through the cervix (Figs 3 and 4). During the morcellation procedure, a protective barrier was wrapped around the uterus and was used for the isolation and prevention of a potential intraperitoneal visceral injury (Fig 5A and 5B). In the future, during the in-vivo procedure, a proximity magnetic

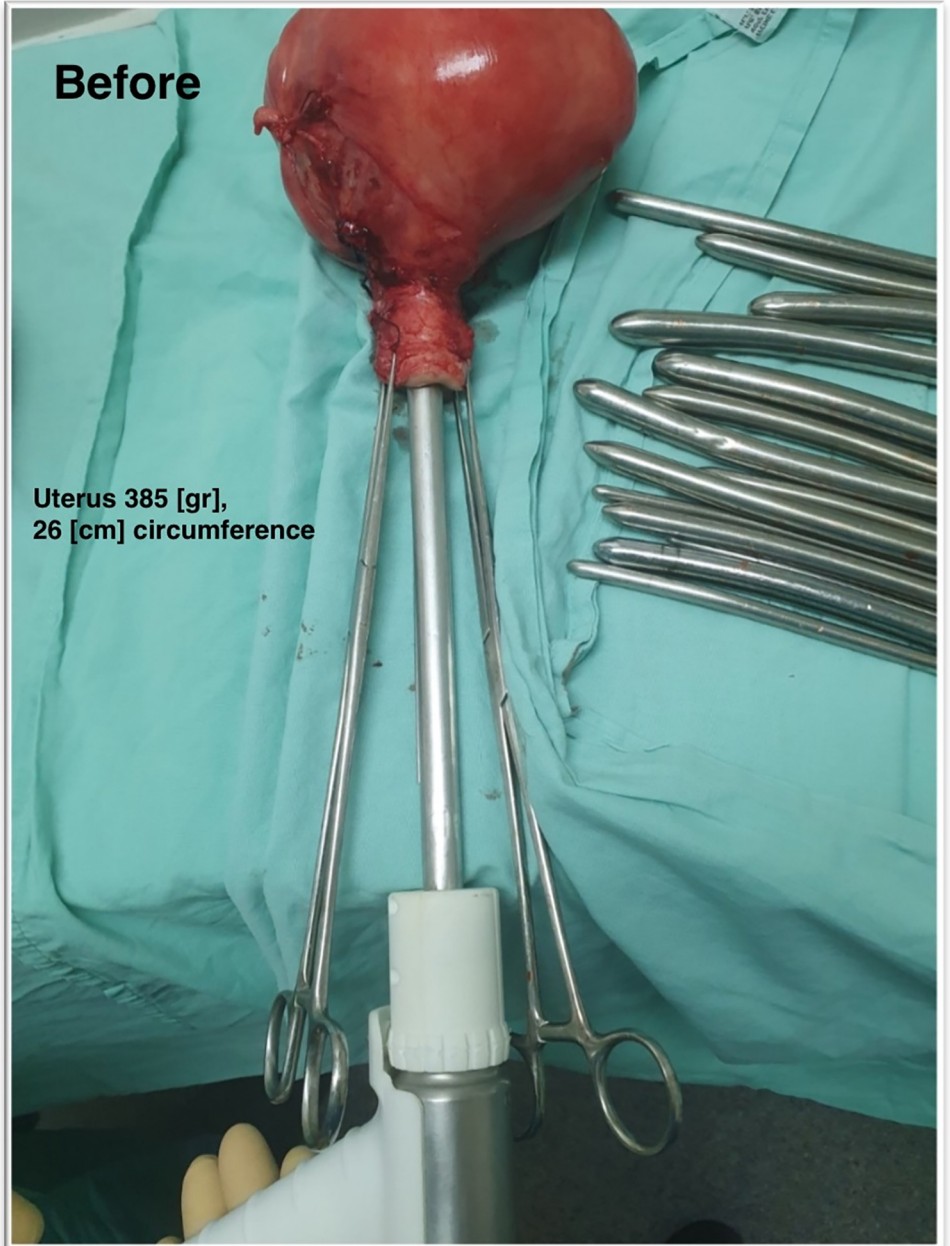

**Fig 3. The Heracure resection device insertion through the cervix.** (Fig 3, Republished from Heracure CP-0300 under a CC BY license, with permission from Heracure company, original copyright 2023).

sensor located inside the protective barrier, will automatically stop the morcellation motor, when the morcellator knives are close to the barrier, to prevent any iatrogenic damage. A laparoscopic camera will accompany the procedure to verify safety and prevent potential visceral injury.

The morcellation procedures were performed by the surgical gynecologist (principal investigator), and the resection time was recorded.

Upon completion of the resection procedure, the reduced weight and size of extirpated uterus was recorded. Post-procedure included testing the uterus for leakage: in case of intact

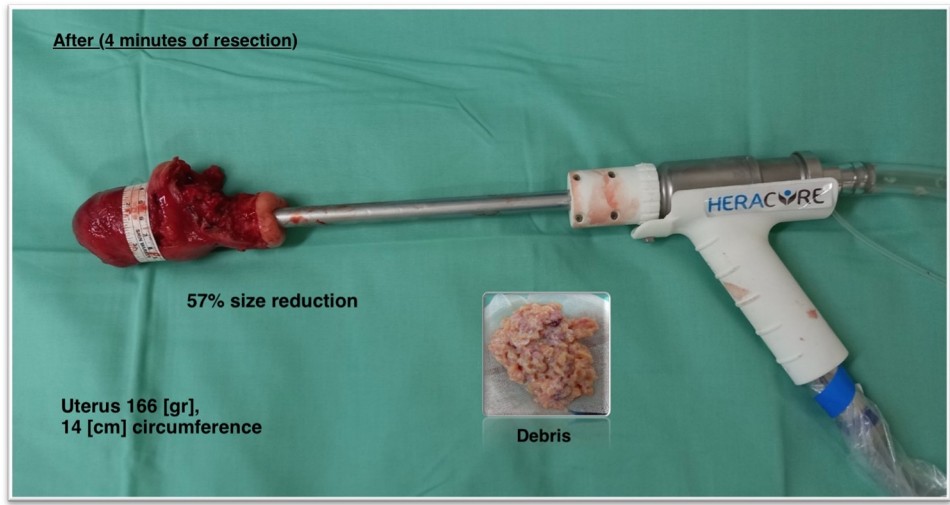

**Fig 4. Resected uterine mass.** (Fig 4, Republished from Heracure CP-0300 under a CC BY license, with permission from Heracure company, original copyright 2023).

fallopian tubes, the tubes were sealed, and uterine cavity was fully filled with saline using a syringe and the outer surface of the uterus was visually observed for leakage (Fig 6). Following the completion of the leakage testing, the extirpated uterus and resected tissue (debris) were collected and sent for routine pathology.

## Statistical analysis

Nominal data were described as numbers and percentages. Continuous data were assessed for normal distribution (Shapiro-Wilk test) and were described as mean ± SD or median (minimum-maximum). All analyses were performed using SPSS-26 software (IBM, Armonk, NY, USA).

## Results

Ten patients were screened for eligibility, and all were found eligible. Ten uteri were examined in this trial. No major complications occurred during the procedure. During the study period, various uteri sizes were selected. Mean pre-procedure uterus size was 290 gr. (weight range: 40–793 gr.; 12–36 cm/ circumference). All ten (10) uteri were successfully reduced in size, (mean uterus weight reduction was 21% with mean circumference reduction of 25% (reduction range: 9% - 54%)). The mean resection time using Heracure device was 4.3 minutes (range: 1– 10min). Table 1 presents an overview of the patient demographic and surgical data.

No leakage was observed from the outer surface of the uterus/serosa at the saline injection post-procedure examination. All uteri were benign upon pathological examination.

Patient list and peri-operative data are presented in Table 2.

## Discussion

In the current study we verified the efficacy and safety of the Heracure device for vaginal intra-uterine morcellation by an extra-uterine model. This technique was developed to overcome the challenge following hysterectomy of large size leiomyomatous uteri, to enable rapid and easy vaginal removal and specimen retrieval that require fragmentation or morcellation [20]. Thus, these larger sized uteri (up to 793 gr, 36 cm/circumference) selected in the current study

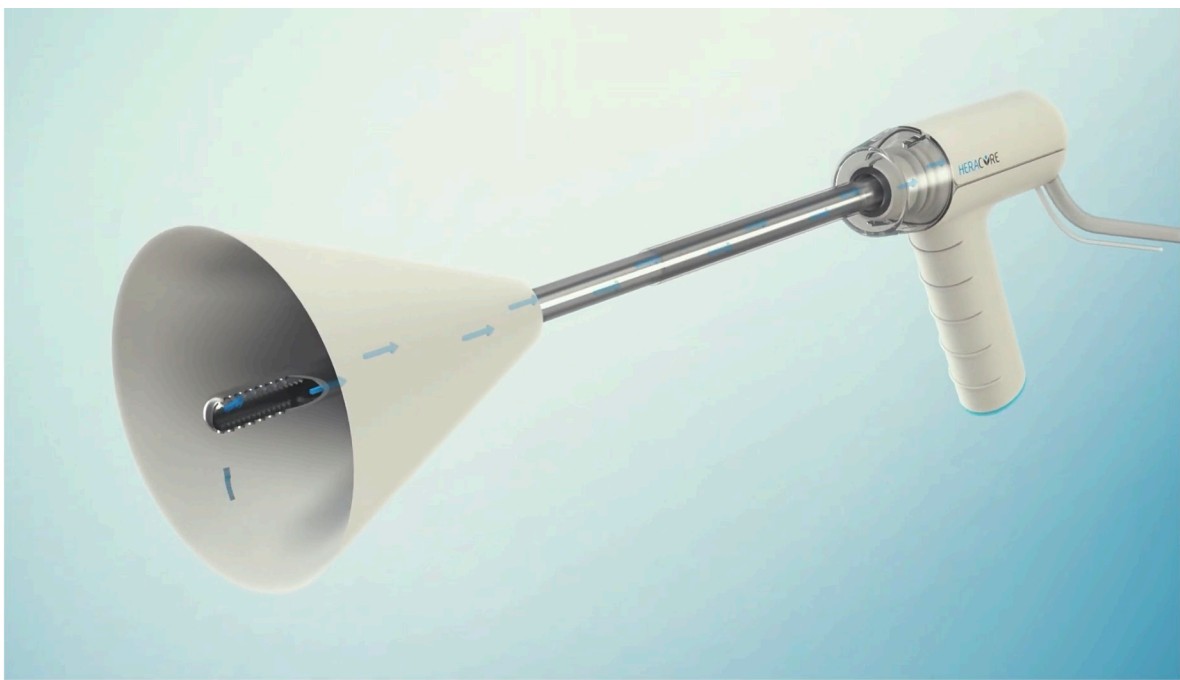

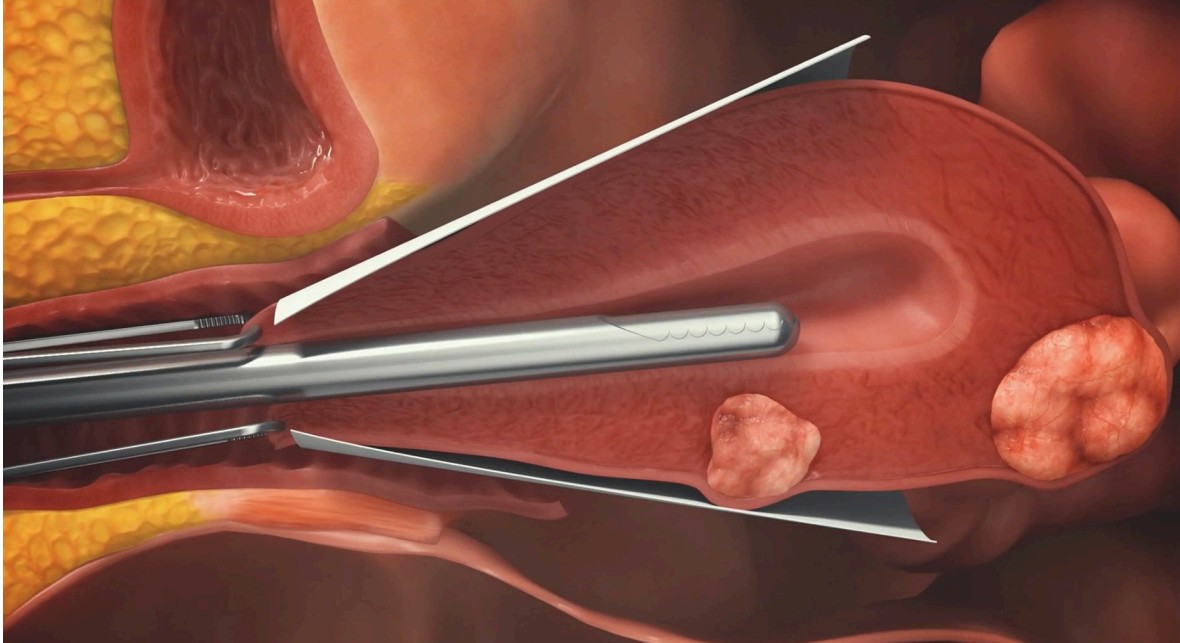

**Fig 5. A and B: Protective cover wrapped around the uterus for the isolation and prevention of potential intraperitoneal visceral injury.** (Fig 5A and 5B, Republished from Heracure CP-0300 under a CC BY license, with permission from Heracure company, original copyright 2023).

were obviously a better fit to demonstrate the efficacy of the Heracure device. Following morcellation procedure with the Heracure device, we demonstrated a mean uterus weight reduction of 21% with mean circumference reduction of 25% from original uterine size. Mean procedure length was 4.3 min. Procedural safety was high, as no cases of uterine perforation occurred.

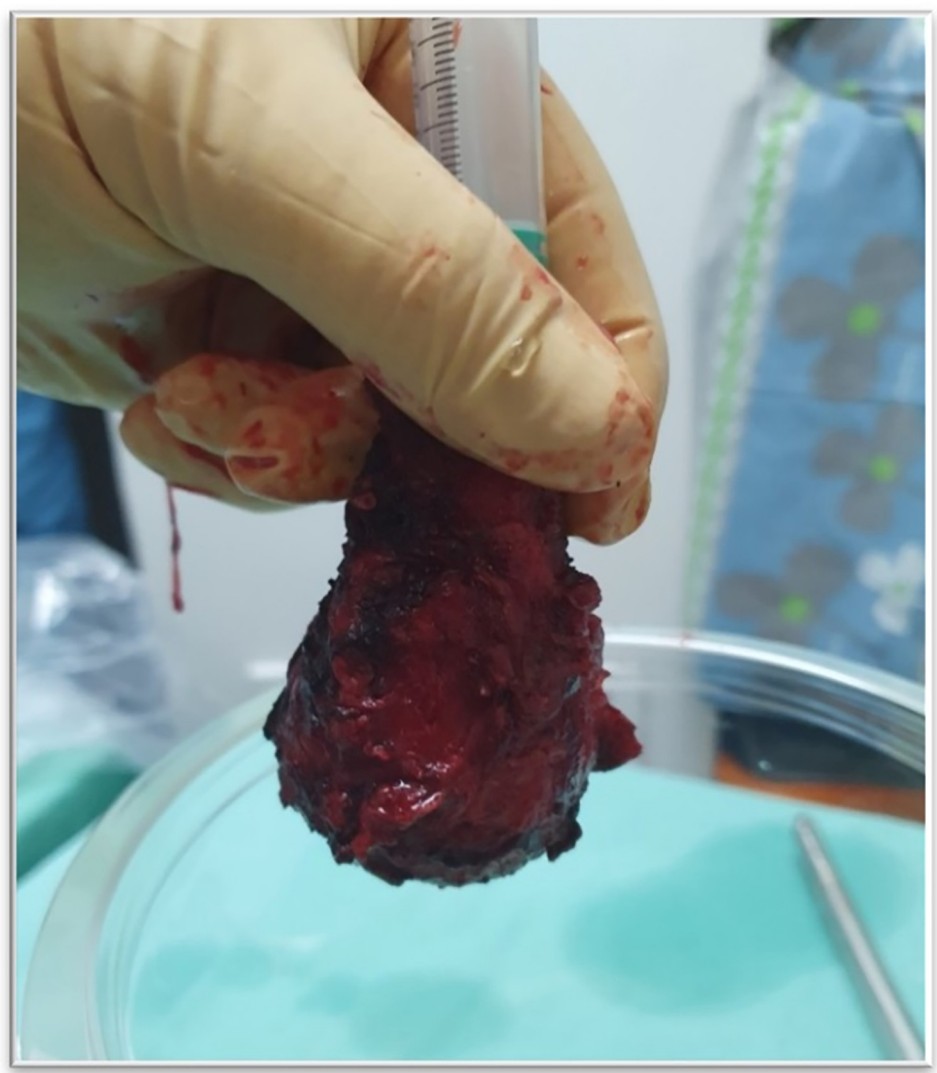

**Fig 6. Leakage test: The uterine cavity was filled with saline using a 10cc syringe.** (Fig 6, Republished from Heracure CP-0300 under a CC BY license, with permission from Heracure company, original copyright 2023).

**Table 1. Patient demographic and perioperative characteristics.**

| Variables | Mean | Range (min-max) |
|---|---|---|
| Age (years) | 57 | 42–77 |
| BMI (kg/m2) | 26 | 21–30 |
| Pre-resection uterus weight (g) | 290 | 40–793 |
| Total resection time (min) | 4.3 | 1.0–10.2 |
| Uterus–Weight Reduction (%) | 21.6 | 9.2–53.6 |
| Uterus–Circumference Reduction (%) | 24.5 | 8.3–46.2 |

BMI: body mass index.

**Table 2. Patient list and perioperative data.**

| Patient | Age [years] | BMI | Route of hysterectomy | Indication for hysterectomy | Pre-resection Uterus weight [g] | Pre-resection Uterus Circumference [cm] | Resection Time [Min] | Post-resection Uterus weight [g] | Post-resection Uterus Circumference [cm] | Uterus–Weight Reduction (%) | Uterus–Circumference Reduction (%) |
|---|---|---|---|---|---|---|---|---|---|---|---|
| 1 | 61 | 22 | LH | Risk reduction* | 358 | 26 | 4.1 | 166 | 14 | 53.6 | 46.2 |
| 2 | 71 | 29 | LH | Fibroid & PMB | 66 | 12 | 1.2 | 57 | 10 | 13.6 | 16.7 |
| 3 | 48 | 21 | LH | Fibroid 16W | 146 | 18.9 | 6.2 | 130 | 13 | 11.0 | 31.2 |
| 4 | 77 | 28 | VH | Prolapse | 40 | 10 | 2.1 | 33 | 8 | 17.5 | 20.0 |
| 5 | 42 | 29 | TAH | Fibroid 18W | 793 | 36 | 10.2 | 720 | 33 | 9.2 | 8.3 |
| 6 | 66 | 25 | LAVH | Prolapse | 63 | 12 | 1.1 | 47 | 8 | 25.4 | 33.3 |
| 7 | 66 | 30 | V Notes | Prolapse | 85 | 13.5 | 1.0 | 73 | 8 | 14.1 | 40.7 |
| 8 | 45 | 23 | LH | Endometriosis | 94 | 15 | 1.2 | 77 | 13 | 18.1 | 13.3 |
| 9 | 49 | 23 | TAH | Fibroid 19W | 531 | 30 | 7.0 | 476 | 27 | 10.4 | 10.0 |
| 10 | 49 | 27 | TAH | Fibroid 17W | 724 | 35 | 9.5 | 487 | 26 | 32.7 | 25.7 |

BMI: body mass index, W: weeks, VH: Vaginal hysterectomy, LH: Laparoscopic hysterectomy, LAVH: Laparoscopic assisted vaginal hysterectomy, TAH: Total abdominal hysterectomy, PMB: Post-menopausal bleeding.

*Lynch syndrome.

Minimally invasive gynecologic surgery often necessitates employing tissue extraction techniques to remove a large specimen through a small incision. Specimen extraction after hysterectomy is a major challenge, thus intraperitoneal, electromechanical, or cold knife morcellation are a mandatory step to benefit from the advantages of the minimally invasive approach.

Two recent studies [21, 22] found that the strongest pre-operative predictors Relative Risk (RR) for the need of morcellation were the leiomyoma diameter of >40–44 mm (RR: 3.58), uterine cross-sectional area of >36.5–48.6 cm$^2$ (RR: 6.38), uterine size in the bimanual exam of >11.5–13 weeks pregnancy (RR: 3.57) and uterine volume of 262 ml. Highlighting, that during the selection of the surgical approach, uterine size plays a major and foremost decisive factor, over patient or surgeon's preference.

To date, the preferred technique to perform a hysterectomy is via conventional vaginal surgery. In cases vaginal hysterectomy is not feasible, a laparoscopic hysterectomy should be offered, to avoid the need for an abdominal hysterectomy [23]. It is well known that uterine size is the main limitative decisive factor in surgical approach selection with a cutoff of 12–18 weeks uterine size indicating vaginal or laparoscopic approach [24]. Therefore, the Heracure intrauterine morcellation device can be the key solution, in such cases that require removal of enlarged myomatous uteri, predominately, to preserve the vaginal hysterectomy approach.

In 2019, the American College of Obstetricians and Gynecologists (ACOG) [25] published a committee opinion regarding the advantages and the risks of intra-abdominal morcellation. There is a well-accepted consensus on the need for a shared decision-making process, that includes an informed consent explaining the risks and benefits of each surgical approach for presumed leiomyomas. Although an abdominal hysterectomy or myomectomy may reduce the risk of spreading cancer cells, it should be noted that unexpected leiomyosarcomas are found in fewer than 1–13 of every 10,000 surgeries performed for symptomatic leiomyomas [26]. Moreover, in light of this concern, a shift towards performing gynecological surgeries in the abdominal approach was associated with increased morbidity, as compared with minimally invasive approaches [27].

The morcellation technique can be divided into two main approaches: The extraperitoneal and the intraperitoneal. The second subdivision is according to the type of morcellation equipment. Manual morcellation using a scalpel is typically employed to remove a bulky uterine specimen through the mini-laparotomy incision port or more often preferred, during vaginal hysterectomy, through the vaginal route with its natural orifice. As opposed to manual morcellation, electromechanical morcellation is usually employed only in laparoscopic procedures (intraperitoneal morcellation). These methods are comparable in respect to complication rate, albeit manual morcellation via mini laparotomy approach is associated with shorter operative time [28, 29].

When comparing the electrical instrumentation versus morcellation with the scalpel, we can reveal one of the main limitations of the current in-bag morcellation concerning the pathology examination. The electric morcellator involves a mass being reduced, delivered to the pathologist as a slice of ground tissue. Unfortunately, the pathologist is often unable to distinguish the anatomy to be analyzed and to indicate exactly the parts to be analyzed, given that there are no reference points [30]. Conversely, manual morcellation with a scalpel allows the organ or mass to be removed and reconstructed with a good approximation using sutures. This novel tool overcomes this disadvantage providing both the anatomy preservation, such as in cold knife morcellation and the efficacy of the electrical morcellation.

A 2018 study by Ghezzi *et al*. [31] described this technique and reported on beneficial perioperative outcome of transvaginal contained extraction of a surgical specimen in laparoscopic myomectomy. Their study emphasized that this technique represented a valuable minimally invasive alternative to the intracorporal morcellation. In addition, the same group claimed in another publication that surgical outcomes in patients undergoing laparoscopic myomectomy, when comparing power morcellation to transvaginal specimen extraction, the latter demonstrated a reduced need for supplementary rescue analgesia and according to the patients, better cosmetic results [32].

Currently, the morcellation method is not without flaws. The use of intraperitoneal morcellation raises several concerns, such as the dissemination and spread of cancer cell tissues, which can potentially lead to worsened prognosis [33]. Another substantial disadvantage is surgical complication, due to the need for extra and larger ports for access of the intraperitoneal morcellation, in cases of laparoscopic hysterectomy. Additionally, various technical difficulties need to be overcome, such as instrument collision during the morcellation process, and reduced traction of tissue that requires additional entry portals [34, 35]. Our assumption is that the intrauterine vaginal approach may overcome these difficulties by obviating the need for an additional portal entry.

Intrauterine morcellation is also familiar in the hysteroscopy field. The hysteroscopic tissue removal system seems a feasible surgical option regarding operative time and complications [36]. This hysteroscopic intrauterine morcellation was not affected by the FDA warning about morcellation in relation to the possible dissemination of misdiagnosed uterine sarcomas [37, 38]. Unfortunately, the type and size of the myoma remain the biggest challenge in such a technique.

Another main notable risk during the morcellation procedure is the potential injury of adjacent intra-abdominal organs. In our device, in addition to direct laparoscopic visualization, the use of a protective barrier/cover (Fig 5A and 5B), wrapped around the uterus has successfully demonstrated to provide another protective shell, to shield adjacent organs (bladder and bowel) from perforation during the procedure- indeed no perforations occurred in this preliminary trial.

This trial demonstrates the feasibility of a novel morcellation approach which can theoretically solve the current day-to-day difficulties in the extraction of large uteruses during a

minimally invasive approach for hysterectomy by introducing a new intrauterine morcellation device. This unique approach enables the extraction of the uterus, even in cases of large myomas that currently preclude the use of morcellation in the laparoscopic and vaginal approach [6]. The potential clinical advantage of intrauterine morcellation is avoiding the need for manual vaginal morcellation and its potential risks. Moreover, this tool may reduce the risk of intraperitoneal spreading of potentially malignant cells, as well as overcome some technical difficulties and improve cosmetic results of the traditional approach for extraction of the uterus.

## Strengths

The main strength of this first preliminary report is its novelty of demonstrating the feasibility and safety of a new intrauterine approach to perform morcellation for uterine size reduction. In our opinion, safety, the most critical point at this stage was demonstrated by the ability to avoid perforation. The Heracure device will enable the use of a minimally invasive approach (laparoscopic and vaginal) for hysterectomy in cases that currently are feasible only with vaginal manual morcellation or need to increase the abdominal access, due to the size of the uterus.

## Limitations

The current trial has several limitations. The main limitation was its small cohort size, that included only three cases of larger uteri with a mean size of 18w and mean resection time of 8.8 minutes. To further assess whether this device can be effective in such cases, future in-vivo trials, will involve a higher number of patients with large uteri, to approve its feasibility and superiority over the traditional approaches. An additional limitation is the trial excluded cases with pathologies that involve malignancies. Of note, the current approach, intrauterine morcellation, as opposed to intraperitoneal morcellation, reduces the uterine size without exposure of morcellated tissue into the abdominal cavity.

## Conclusions

In conclusion, the Heracure device is a novel tool for the reduction of the uterine size and may potentially be applied to perform vaginal/ laparoscopic hysterectomy in specific cases of larger volume size uteri, which require size reduction prior to removal.

We hope this novel device will enable more gynecologic surgeons to perform hysterectomies in an easier and more rapid manner, yet with assured safety in cases of limited access, due to large uterine size or limited laparoscopic approach. Future in-vivo trials are warranted to strengthen the findings of this limited observational cohort for the safety and efficacy of the procedure.

## Supporting information

**S1 File.**
(PDF)

## Acknowledgments

**Paper presentation:** This trial was accepted for presentation at the ESGE (European Society for Gynecological Endoscopy) 31st Annual Congress, October 2022, Lisbon, Portugal.

## Author Contributions

**Conceptualization:** Meir Pomeranz, Yair Daykan.

**Data curation:** Meir Pomeranz.

**Methodology:** Meir Pomeranz, Ron Schonman, Zvi Klein.

**Resources:** Meir Pomeranz.

**Supervision:** Meir Pomeranz, Ron Schonman, Zvi Klein.

**Writing – original draft:** Yair Daykan.

**Writing – review & editing:** Meir Pomeranz, Ron Schonman, Yael Yagur, Rina Tamir Yaniv, Zvi Klein, Yair Daykan.

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
