## [Decision Letter · Decision Letter 0]

1 Nov 2022

PONE-D-22-26314An Intrauterine Cavity Morcellator: A Novel Approach to High Volume Uterus Morcellation. EX-VIVO StudyPLOS ONE

Dear Dr. Daykan,

Thank you for submitting your manuscript to PLOS ONE. After careful consideration, we feel that it has merit but does not fully meet PLOS ONE’s publication criteria as it currently stands. Therefore, we invite you to submit a revised version of the manuscript that addresses the points raised during the review process.

We look forward to receiving your revised manuscript.

Kind regards,

Andrea Giannini

Academic Editor

PLOS ONE

Journal Requirements:

3. PLOS requires an ORCID iD for the corresponding author in Editorial Manager on papers submitted after December 6th, 2016. Please ensure that you have an ORCID iD and that it is validated in Editorial Manager. To do this, go to ‘Update my Information’ (in the upper left-hand corner of the main menu), and click on the Fetch/Validate link next to the ORCID field. This will take you to the ORCID site and allow you to create a new iD or authenticate a pre-existing iD in Editorial Manager. Please see the following video for instructions on linking an ORCID iD to your Editorial Manager account: https://www.youtube.com/watch?v=_xcclfuvtxQ.

6. Please ensure that you refer to Figure 1 in your text as, if accepted, production will need this reference to link the reader to the figure.

7. We note that the original protocol that you have uploaded as a Supporting Information file contains an institutional logo. As this logo is likely copyrighted, we ask that you please remove it from this file and upload an updated version upon resubmission.

8. We note that the original protocol file you uploaded contains a confidentiality notice indicating that the protocol may not be shared publicly or be published. Please note, however, that the PLOS Editorial Policy requires that the original protocol be published alongside your manuscript in the event of acceptance. Please note that should your paper be accepted, all content including the protocol will be published under the Creative Commons Attribution (CC BY) 4.0 license, which means that it will be freely available online, and any third party is permitted to access, download, copy, distribute, and use these materials in any way, even commercially, with proper attribution.

Therefore, we ask that you please seek permission from the study sponsor or body imposing the restriction on sharing this document to publish this protocol under CC BY 4.0 if your work is accepted. We kindly ask that you upload a formal statement signed by an institutional representative clarifying whether you will be able to comply with this policy. Additionally, please upload a clean copy of the protocol with the confidentiality notice (and any copyrighted institutional logos or signatures) removed.

Additional Editor Comments:

Dear authors,

the topic of the present article titled “An Intrauterine Cavity Morcellator: A Novel Approach to High Volume Uterus Morcellation. EX-VIVO Study” is very interesting, the paper and the aim falls within the scope of the journal but the article needs major improvements.

The introduction, material and method section and tables should be modified and improved.

The manuscript should be organized better and English should be improved.

I suggest improving the manuscript with the reviewers' comments.

Reviewers' comments:

Reviewer's Responses to Questions

**Comments to the Author**

1. Is the manuscript technically sound, and do the data support the conclusions?

Reviewer #1: Yes

Reviewer #2: Yes

Reviewer #3: No

Reviewer #4: Yes

Reviewer #5: Partly

Reviewer #6: Partly

2. Has the statistical analysis been performed appropriately and rigorously? 

Reviewer #1: Yes

Reviewer #2: Yes

Reviewer #3: N/A

Reviewer #4: Yes

Reviewer #5: I Don't Know

Reviewer #6: No

3. Have the authors made all data underlying the findings in their manuscript fully available?

Reviewer #1: Yes

Reviewer #2: Yes

Reviewer #3: Yes

Reviewer #4: Yes

Reviewer #5: No

Reviewer #6: Yes

4. Is the manuscript presented in an intelligible fashion and written in standard English?

Reviewer #1: Yes

Reviewer #2: Yes

Reviewer #3: Yes

Reviewer #4: Yes

Reviewer #5: Yes

Reviewer #6: Yes

5. Review Comments to the Author

Reviewer #1: This study gives an insight to successful use of intra uterine morcellator.

There are a few spelling errors needs to be corrected.

The author mentions use of t test and p value in the methodology whereas there has been no such analysis in the results section

Reviewer #2: I read with great interest the Manuscript titled “An Intrauterine Cavity Morcellator: A Novel Approach to High Volume Uterus Morcellation. EX-VIVO Study” (PONE-D-22-26314), which falls whithin the aim of Plos One.

In my honest opinion, the topic is interesting enough to attract the readers’ attention. Methodology is accurate and conclusions are supported by the data analysis. Nevertheless, authors should clarify some point and improve the discussion citing relevant and novel key articles about the topic.

Authors should consider the following recommendations:

- Manuscript should be further revised by a native English speaker.

- The Authors did not mention the sample size calculation for their study. It is essential to specify this data in order to guarantee an adequate significance of the results obtained by the Authors.

- The authors have not adequately highlighted the strengths and limitations of their study. I suggest better specifying these points.

- What are the actual clinical implications of this study? it is important to report the results obtained by the authors in the context of clinical practice and to adequately highlight what contribution this study adds to the literature already existing on the topic and to future study perspectives.

- Does this manuscript conform the Enhancing the QUAlity and Transparency Of health Research (EQUATOR) network guidelines? It would be mandatory to declare about this element.

Reviewer #3: Power morcellators are usually so useful to pick the removed uterus from abdominal cavity. FDA mentioned that under manufacturer’s indication hysteroscopic morcellator does not raise the risk of spreading cancer or fibroid, which is distinct from laparoscopic open morcellation.

In the present article, there are several points which should be clarified.

#1. Does Heracure morcellator stop automatically or with surgeon’s manipulation?

#2. In general, the uterus, which is too large to remove through vaginal orifice is a good indication for morcellation. However, the uterus removed by TAH (case5, 9and 10) tend to suggest poorer response for morcellation and longer time to spend it. Only from the data suggested in this manuscript, it is uncertain whether Heracure morcellator is more useful compared to other kinds of hysteroscopic morcellators.

Reviewer #4: Dear Author

The topic is current and interesting. The article examines the effectiveness of a morcellator in reducing uterine size intrauterine. As a result, the morcellator used significantly reduced the size of the uterus. Since the study was performed ex-vivo, information about complications could not be obtained. In addition, the number of cases is limited. I think that the study can be published in its current form in order to investigate its use in operations in the future.

Yours sincerely

Reviewer #5: Reviewer comments on the manuscript titled

An Intrauterine Cavity Morcellator: A Novel Approach to High Volume Uterus

Morcellation. EX-VIVO Study

1. In the material and method section Page 10 Lines 84 & 85, you mentioned that Patients were followed-up for 6 weeks post – operative. What is the indication for postoperative follow-up while your technique was done ex-vivo on extirpated uteri? Especially that No intervention with your new technique were applied in the patients. And you mentioned that in the lines 87&88 that (No intervention was applied in the patients, as the study only investigated the post-hysterectomy uterus.)

2. In the page 10 line 90 you mentioned that the primary outcome evaluated was the procedure’s safety, how could be determined by this small number of patients (10 patients) and the ex-vivo procedure and choosing only uteri with benign lesions.

3. In the case selection section, page 11, you selected only uteri with benign lesions, you ought to utilize uteri with benign and malignant lesions where the liability for perforation of uterine walls and spreading of malignant cells is more with malignant uterine lesions where the point of your review is to test the safety of the technique in keeping away from perforating the uterine walls and expected spread of malignant cells or injury to nearby vital organs.

4. The protective barrier that was folded over the uterus and was utilized for the separation and avoidance of potential intraperitoneal visceral injury is opened at its upper end, so won't forestall the spillage to the peritoneal cavity assuming perforation of uterine wall happened especially with the utilization of irrigation fluid in controversy to the use of in bag power morcellation that were shown in previous studies.

5. What is the material and length and flexibility of the barrier used and the method of its intraoperative application to surround the whole uterus especially if it is large volume (as 20 weeks size) with interstitial or subserous myomata?

6. In the case number 3& 5&9 &10 the uterine size was large with fibroids what is the type of it, subserous, submucous or interstitial in each case.

7. In case number 1 what is meant by the indication of risk reduction in a uterus weight 365 gram. What is the pathological diagnosis before and after hysterectomy of this case?

8. As the authors mentioned, (in page 19 line 290), In-VIVO trials are required to strength these finding of this limited observation cohort for the safety and efficacy of the procedure, especially that the application of the protective barrier might limit the manipulation of the uterus and visualization of the surrounding bladder or intestine.

9. The chance of uterine perforation won't diminish with this maneuver since morcellation is done indiscriminately without hysteroscopic guide and the role of laparoscopy is just to screen in the event that perforation happened or not and to protect intestine or bladder from injury.

10. If the uterus enlarged by huge subserous myoma uterine perforation definitely occurs and possibility of dissemination of malignant cells is high than enclosed abdominal morcellation.

Reviewer #6: The study entitled ‘An Intrauterine Cavity Morcellator: A Novel Approach to High Volume Uterus Morcellation. EX-VIVO Study with the aim to evaluate the safety and efficacy of the intra uterine morcellation device during laparoscopic/vaginal hysterectomy.

The manuscript could be improved based on the comments below.

Introduction

Line 5 rapid recovery[1–3], the cited reference in text to be spaced out from the word. This applies throughout the manuscript.

Materials and Methods

Line 87 & Line 130, Ex VIVO is to be written as Ex vivo.

Line 136 & Line 290, In VIVO is to be replaced with In vivo.

Results

Table 1 range is to be denoted as min – max values in the table footnote. Percentage figures to be at least 1 decimal point. Sample size is to be indicated in the table. The word Data Mean Range is to be removed and replaced with Variables. Pre-resection uterine size or uterine weight? Size and weight are different in meaning. The short form for gram is g.

Table 1 & 2, decimal points for minutes to be added. Was the resection time captured in minutes converted from total seconds?

Table 2, columns and data for post-resection uterus size and post-resection uterus circumference to be added before deriving uterus weight and circumference reduction data. Percentage figures to be at least 1 decimal point.

Statistical analyses were stated in the manuscript, but there was no statistical analysis output presented. If there were no statistical tests performed, the writeup on statistical analyses is to be omitted, and the reason is to be stated in the method section.

Line 215, RR is to be spelled out when first used.

Discussion

Line 216, proper unit is to be presented for cm2 e.g cm ^2 (superscript 2).

Some page numbers which were provided in the TREND Checklist did not tally with the page number in the manuscript. If there is no information provided in the manuscript, the page number should not to be written in the checklist. A write-up could be provided to mention those information were not relevant/applicable in the manuscript.

6. PLOS authors have the option to publish the peer review history of their article (what does this mean?). If published, this will include your full peer review and any attached files.

Reviewer #1: **Yes: **Shreyashi Aryal

Reviewer #2: No

Reviewer #3: No

Reviewer #4: **Yes: **Remzi Atilgan

Reviewer #5: **Yes: **Mohamed Ibrahim Mohamed Amer

Reviewer #6: No

---

## [Author Response · Author response to Decision Letter 0]

28 Nov 2022

28.11.2022

Dr. Andrea Giannini

Academic Editor

PLOS ONE

Re: Manuscript ID PONE-D-22-26314

An Intrauterine Cavity Morcellator: A Novel Approach to High Volume Uterus Morcellation. EX-VIVO Study

Dear Dr.Giannini

Thank you for your letter regarding our above-cited manuscript. We would also like to thank the reviewers for their thoughtful comments and appropriate suggestions. We have revised the manuscript extensively, carefully addressing all the comments and suggestions. The changes made in the revised manuscript are indicated by “highlight function" in Word. Our point-by-point responses to all the comments are listed below.

We hope you will now find this manuscript suitable for publication in the PLOS ONE journal.

On behalf of all authors,

Yair Daykan, MD 

Corresponding author

Reviewer(s)' Comments to Author:

Journal Requirements:

Reviewer #1: 

1. There are a few spelling errors needs to be corrected

• Spelling errors and English grammar were corrected

2. The author mentions use of t test and p value in the methodology whereas there has been no such analysis in the results section

• The result section has been corrected.

Reviewer #2: 

1. The Authors did not mention the sample size calculation for their study. It is essential to specify this data in order to guarantee an adequate significance of the results obtained by the Authors.

• Thank you for your question. The study's primary endpoint was to evaluate the feasibility and safety of novel morcellation equipment. Though this was a small-scale pilot study, we couldn’t calculate a sample size for this trial as we don’t have the frequency of perforation to such an intrauterine device. In our subsequent trial, which will be a larger cohort, we will calculate a sample size according to the primary outcome 

2. The authors have not adequately highlighted the strengths and limitations of their study. I suggest better specifying these points.

• Thank you, we elaborate on these at the end of the discussion section.

• Line 273

• " The main strength of this first preliminary report is its novelty of demonstrating the feasibility and safety of a new intrauterine approach to perform morcellation for uterine size reduction. In our opinion, safety, the most critical point at this stage was demonstrated by the ability to avoid perforation. The Heracure device will enable the use of a minimally invasive approach (laparoscopic and vaginal) for hysterectomy in cases that currently are feasible only with vaginal manual morcellation or need to increase the abdominal access due to the size of the uterus. "

3. What are the actual clinical implications of this study? it is important to report the results obtained by the authors in the context of clinical practice and to adequately highlight what contribution this study adds to the literature already existing on the topic and to future study perspectives.

• We elaborate on this point in the discussion section.

• Line 263

• "This trial demonstrates feasibility of a novel morcellation approach, which can theoretically solve the current day-to-day difficulties in the extraction of large uteruses, during a minimally invasive approach for hysterectomy, by introducing a new intrauterine morcellation device. This unique approach enables the extraction of the uterus, even in cases of large myomas that currently preclude the use of the laparoscopic and the vaginal approach morcellation [1,4,5]. The potential clinical advantage of intrauterine morcellation is avoiding the need for manual vaginal morcellation and its potential risks. Moreover, this tool may reduce the risk of intraperitoneal spreading of potentially malignant cells as well as overcome some technical difficulties and improve cosmetic results of the traditional approach for extraction of the uterus."

4. Does this manuscript conform the Enhancing the Quality and Transparency Of health Research (EQUATOR) network guidelines? It would be mandatory to declare about this element.

• We declare this element in the methods section.

• Line 96

• " This study used the implementation of enhancing the Quality and Transparency Of health Research (EQUATOR) network guidelines."

Reviewer #3: 

1. Does Heracure morcellator stop automatically or with surgeon’s manipulation?

• We elaborate on this important point in the text.

• Line 128

• " In the future during the in-vivo procedure, a proximity magnetic sensor located inside the protective barrier will automatically stop the morcellation motor when the morcellator knives are close to the barrier, to prevent any iatrogenic damage. A laparoscopic camera will accompany the procedure to verify safety and prevent potential visceral injury."

2. In general, the uterus, which is too large to remove through vaginal orifice is a good indication for morcellation. However, the uterus removed by TAH (case5, 9and 10) tend to suggest poorer response for morcellation and longer time to spend it. Only from the data suggested in this manuscript, it is uncertain whether Heracure morcellator is more useful compared to other kinds of hysteroscopic morcellators.

• Thank you for this important question. The TAH morcellation cases had enlarged uteruses with a mean size of 18w and a mean resection time of 8.8 minutes. In this preliminary study, we aim to assess the feasibility of this device and included only three cases of such large uteri. Future in-vivo trials will involve a higher number of patients with large uteri to approve its feasibility and superiority over the traditional approaches. Moreover, the goal of this new intrauterine device is morcellation for uterine size reduction during hysterectomy. As far as we know, the use of current hysteroscopic morcellators are for intrauterine pathologies when uterine preservation is needed. As these are two different goals of treatment they are not comparable. 

• We added this clarification to the limitation section. 

• Line 282

• " The current trial has several limitations. The main limitation was its small cohort size that included only three cases of larger uteri with a mean size of 18w and mean resection time of 8.8 minutes. To further assess whether this device can be effective in such cases, future in-vivo trials, will involve a higher number of patients with large uteri, to approve its feasibility and superiority over the traditional approaches."

Reviewer #5: 

1. In the material and method section Page 10 Lines 84 & 85, you mentioned that Patients were followed-up for 6 weeks post – operative. What is the indication for postoperative follow-up while your technique was done ex-vivo on extirpated uteri? Especially that No intervention with your new technique were applied in the patients. And you mentioned that in the lines 87&88 that (No intervention was applied in the patients, as the study only investigated the post-hysterectomy uterus.)

• Thank you for this comment, we accept it, and since this study investigates only the post-hysterectomy uterus, we omitted this sentence.

2. In the page 10 line 90 you mentioned that the primary outcome evaluated was the procedure’s safety, how could be determined by this small number of patients (10 patients) and the ex-vivo procedure and choosing only uteri with benign lesions.

• In every innovative medical tool, it is accepted to evaluate first using a small number of patients to prove the concept and safety. In our opinion, safety was the most critical point at this stage and was expressed by the ability to avoid perforation. We were not aiming for cases with malignancies (part of the exclusion criteria) and only focused on benign lesions such as myomas or cases of prolapsed uteri. Usually, those pathologies rarely involve malignancies. Furthermore, intrauterine morcellation reduces uterine size without exposure of morcellated tissue to the abdominal cavity. 

• Line 282

• " The current trial has several limitations. The main limitation was its small cohort size… 

An additional limitation is the trial excluded cases with pathologies that involve malignancies. Of note, the current approach, intrauterine morcellation, as opposed to intraperitoneal morcellation, reduces the uterine size without exposure of morcellated tissue into the abdominal cavity. "

3. In the case selection section, page 11, you selected only uteri with benign lesions, you ought to utilize uteri with benign and malignant lesions where the liability for perforation of uterine walls and spreading of malignant cells is more with malignant uterine lesions where the point of your review is to test the safety of the technique in keeping away from perforating the uterine walls and expected spread of malignant cells or injury to nearby vital organs.

The protective barrier that was folded over the uterus and was utilized for the separation and avoidance of potential intraperitoneal visceral injury is opened at its upper end, so won't forestall the spillage to the peritoneal cavity assuming perforation of uterine wall happened especially with the utilization of irrigation fluid in controversy to the use of in bag power morcellation that were shown in previous studies.What is the material and length and flexibility of the barrier used and the method of its intraoperative application to surround the whole uterus especially if it is large volume (as 20 weeks size) with interstitial or subserous myomata?

• Thank you for this question- the barrier is flexible and arrives in different sizes to adjust different sizes of uterine. We elaborate further on these important points in the text.

• Line 128

• " In the future, during the in-vivo procedure, a proximity magnetic sensor located inside the protective barrier will automatically stop the morcellation motor when the morcellator knives are close to the barrier, to prevent any iatrogenic damage. A laparoscopic camera will accompany the procedure to verify safety and prevent potential visceral injury."

4. In the case number 3& 5&9 &10 the uterine size was large with fibroids what is the type of it, subserous, submucous or interstitial in each case.

• In all the myoma cases, we had type 1-5 myoma (FIGO classification.)

- Case 3- Type 4

- Case 5- Type 1,3,5

- Case 9- Type 5

- Case 10-Type 4-5

5. In case number 1 what is meant by the indication of risk reduction in a uterus weight 365 gram. What is the pathological diagnosis before and after hysterectomy of this case?

• This woman had Lynch syndrome- It was mentioned in Table 2 

8. As the authors mentioned, (in page 19 line 290), In-VIVO trials are required to strength these finding of this limited observation cohort for the safety and efficacy of the procedure, especially that the application of the protective barrier might limit the manipulation of the uterus and visualization of the surrounding bladder or intestine.The chance of uterine perforation won't diminish with this maneuver since morcellation is done indiscriminately without hysteroscopic guide and the role of laparoscopy is just to screen in the event that perforation happened or not and to protect intestine or bladder from injury.If the uterus enlarged by huge subserous myoma uterine perforation definitely occurs and possibility of dissemination of malignant cells is high than enclosed abdominal morcellation.

• The barrier is flexible and arrives in different sizes to adjust different sizes of uterine. In the future, during the in-vivo procedure, a proximity magnetic sensor located inside the protective barrier will automatically stop the morcellation motor when the morcellator knives are close to the barrier, to prevent any iatrogenic damage.

Reviewer #6: 

Introduction

1. Line 5 rapid recovery [1–3], the cited reference in text to be spaced out from the word. This applies throughout the manuscript.

• Thank you for this point- we corrected this throughout the manuscript.

Materials and Methods

2. Line 87 & Line 130, Ex VIVO is to be written as Ex vivo.

3. Line 136 & Line 290, In VIVO is to be replaced with In vivo

• This change was modified throughout the manuscript.

Results

4. Table 1 range is to be denoted as min – max values in the table footnote. Percentage figures to be at least 1 decimal point. Sample size is to be indicated in the table. The word Data Mean Range is to be removed and replaced with Variables. Pre-resection uterine size or uterine weight? Size and weight are different in meaning. The short form for gram is g.

• Thank you for this comment; these points were modified in the table

5. Table 1 & 2, decimal points for minutes to be added. Was the resection time captured in minutes converted from total seconds?

• It was added. The time taken in a sec and converted to min. 

• For example: 250 sec /60=4.1 min

6. Table 2, columns and data for post-resection uterus size and post-resection uterus circumference to be added before deriving uterus weight and circumference reduction data. Percentage figures to be at least 1 decimal point.

• We added it to the table.

7. Statistical analyses were stated in the manuscript, but there was no statistical analysis output presented. If there were no statistical tests performed, the writeup on statistical analyses is to be omitted, and the reason is to be stated in the method section.

• We corrected the statistic section.

8. Line 215, RR is to be spelled out when first used

• Was added to the text

• Line 208

Discussion

9. Line 216, proper unit is to be presented for cm2 e.g cm ^2 (superscript 2).

• Was corrected in the text

• Line 210

10. Some page numbers which were provided in the TREND Checklist did not tally with the page number in the manuscript. If there is no information provided in the manuscript, the page number should not to be written in the checklist. A write-up could be provided to mention those information were not relevant/applicable in the manuscript

• The TREND Checklist was corrected. Places without information in the manuscript write-up as NA

---

## [Decision Letter · Decision Letter 1]

27 Dec 2022

PONE-D-22-26314R1An Intrauterine Cavity Morcellator: A Novel Approach to High Volume Uterus Morcellation. EX-VIVO StudyPLOS ONE

Dear Dr. Daykan,

Thank you for submitting your manuscript to PLOS ONE. After careful consideration, we feel that it has merit but does not fully meet PLOS ONE’s publication criteria as it currently stands. Therefore, we invite you to submit a revised version of the manuscript that addresses the points raised during the review process.

We look forward to receiving your revised manuscript.

Kind regards,

Andrea Giannini

Academic Editor

PLOS ONE

Journal Requirements:

Additional Editor Comments :

Dear Authors,

the manuscript it has now been evaluated by our experts and they have recommended that minor changes be made to the submission.

Please improving the manuscript with the reviewers' comments.

Reviewers' comments:

Reviewer's Responses to Questions

**Comments to the Author**

1. If the authors have adequately addressed your comments raised in a previous round of review and you feel that this manuscript is now acceptable for publication, you may indicate that here to bypass the “Comments to the Author” section, enter your conflict of interest statement in the “Confidential to Editor” section, and submit your "Accept" recommendation.

Reviewer #3: All comments have been addressed

Reviewer #4: All comments have been addressed

Reviewer #6: All comments have been addressed

Reviewer #7: All comments have been addressed

Reviewer #8: (No Response)

2. Is the manuscript technically sound, and do the data support the conclusions?

Reviewer #3: Partly

Reviewer #4: Yes

Reviewer #6: Partly

Reviewer #7: Yes

Reviewer #8: Partly

3. Has the statistical analysis been performed appropriately and rigorously? 

Reviewer #3: Yes

Reviewer #4: Yes

Reviewer #6: Yes

Reviewer #7: Yes

Reviewer #8: Yes

4. Have the authors made all data underlying the findings in their manuscript fully available?

Reviewer #3: Yes

Reviewer #4: Yes

Reviewer #6: Yes

Reviewer #7: Yes

Reviewer #8: Yes

5. Is the manuscript presented in an intelligible fashion and written in standard English?

Reviewer #3: Yes

Reviewer #4: Yes

Reviewer #6: Yes

Reviewer #7: Yes

Reviewer #8: Yes

6. Review Comments to the Author

Reviewer #3: Thank you for resubmitting your manuscript.

As I mentioned previously, the entry number is too small. It seems a kind of preliminary data with supporting Heracure Medical Ltd. It must not reach to the level of this journal.

According to your result, the indication to apply Heracure device seems rather small. The enclosed in-bag abdominal morcellation is more useful to remove larger-sized uterus.

You need to analyze not only the weight and circumference of the uterus but also the shape and the level of flexio to regulate the appropriate indication.

In addition, I am wondering whether the radial panel completely protect the dissemination of the small component of the removing uterus. Compare with the protection you suggested in the revised manuscript, in-bag morcellation seems rather simple and safe. I hope you to submit the renewal manuscript with additional information after further investigation.

Reviewer #4: Dear Author, The article has been edited in line with the suggestions. I think the article can be published in its current form.

Reviewer #6: (No Response)

Reviewer #7: In the discussion, it would be appropriate to include the usual problem of morcellation with electrical instrumentation, versus morcellation with the scalpel. The electric morcellator involves mass being reduced by morcellation of tissue which is delivered to the pathologist as a meat "hamburger". The pathologist, unfortunately, is often unable to distinguish the anatomy to be analyzed and to indicate exactly the parts to be analyzed in anatomical reference of the bowel, given that he has no points of reference (PMID: 36292534). Conversely, manual morcellation with the scalpel allows the organ or mass to be removed and reconstructed with a good approximation using sutures. Therefore I am convinced that this specification can be useful to readers of such a complex and controversial topic.

Reviewer #8: I read with great interest the manuscript, which falls within the aim of this Journal. In my honest opinion, the topic is interesting enough to attract the readers’ attention. Nevertheless, the authors should clarify some points and improve the discussion, as suggested below.

Authors should consider the following recommendations:

- Manuscript should be further revised in order to correct some typos and improve style.

- Authors should add further elements to discuss the role and technical differences among Intrauterine Cavity Morcellator (authors may refer to: PMID: 28948169), highlighting that these devices were not affected by the FDA warning about morcellation, which led to the adoption of endogag use as mandatory to avoid the dissemination of misdiagnosed uterine sarcomas (authors may discuss this point referring to: PMID: 35263843; PMID: 36553227).

- I recommend to highlight novel pieces of evidence about the minimally invasive management of hysterectomy for large uteri and/or for different indications, as well as the surgical specimen extraction (authors may refer to: PMID: 36498515; PMID: 34730067).

7. PLOS authors have the option to publish the peer review history of their article (what does this mean?). If published, this will include your full peer review and any attached files.

Reviewer #3: No

Reviewer #4: **Yes: **Remzi Atılgan

Reviewer #6: No

Reviewer #7: **Yes: **ANDREA TINELLI

Reviewer #8: No

---

## [Author Response · Author response to Decision Letter 1]

15 Jan 2023

15.1.2023

Dr. Andrea Giannini

Academic Editor

PLOS ONE

Re: PONE-D-22-26314R1

An Intrauterine Cavity Morcellator: A Novel Approach to High Volume Uterus Morcellation. EX-VIVO Study

Dear Dr.Giannini

Thank you for your letter regarding our above-cited manuscript. We would also like to thank the reviewers for their thoughtful comments and appropriate suggestions. We have revised the manuscript extensively, carefully addressing all the comments and suggestions. The changes made in the revised manuscript are indicated by “highlight function" in Word. Our point-by-point responses to all the comments are listed below.

We hope you will now find this manuscript suitable for publication in the PLOS ONE journal.

On behalf of all authors,

Yair Daykan, MD 

Corresponding author

Reviewer(s)' Comments to Author:

Journal Requirements:

Reviewer #3: 

Thank you for resubmitting your manuscript.

As I mentioned previously, the entry number is too small. It seems a kind of preliminary data with supporting Heracure Medical Ltd. It must not reach to the level of this journal.

According to your result, the indication to apply Heracure device seems rather small. The enclosed in-bag abdominal morcellation is more useful to remove larger-sized uterus.

You need to analyze not only the weight and circumference of the uterus but also the shape and the level of flexion to regulate the appropriate indication.

In addition, I am wondering whether the radial panel completely protect the dissemination of the small component of the removing uterus. Compare with the protection you suggested in the revised manuscript, in-bag morcellation seems rather simple and safe. I hope you to submit the renewal manuscript with additional information after further investigation.

• Dear reviewer, 

Thank you for these comments. The purpose of this manuscript is to present the preliminary experience with a new device in an ex-vivo study. In this preliminary study manuscript, we demonstrate the feasibility of the Heracure device. 

At the initial investigation of every innovative medical tool, it is acceptable to first evaluate proof of concept and safety in a small number of patients. Our subsequent trial in-vivo, will include a larger cohort, in order to demonstrate the advantages of such a tool and highlight the safety with a new added on component to the flexible protective barrier and the proximity magnetic sensor.

We are sure that the readers will also benefit from this publication, despite its small cohort.

Nevertheless, I would like to draw your attention to previous publications that similarly began to investigate an innovative tool like our tool, in a small cohort. For this example, I choose the vNOTES approach.

Source, year Study type No. of patients Main surgery type

Lee et al, 25 2015 Case series 4 Robot-assisted hysterectomy

Wang et al, 18 2016 Retrospective matched case–control 34 NOTES-assisted ovarian cystectomy

Baekelandt, 23 2015, Baekelandt, 24 2016 Case series 10 Hysterectomy

Baekelandt, 29 2015 Video report 1 Adnexectomy

Lee et al, 26 2014 Case series 3 Myomectomy

Lee et al, 27 2014 Case series 3 NOTES-assisted hysterectomy + salpingo-oophorectomy + pelvic lymphadenectomy

Xu et al, 17 2014 Prospective randomized 18 Salpingectomy (pure NOTES = 17, hybrid NOTES = 1)

Yang et al, 21 2014 Retrospective matched case– control 16 NOTES-assisted hysterectomy

Yang et al, 16 2013 Case series 7 Salpingo-oophorectomy( n = 3), oophorectomy ( n = 3), paraovarian cystectomy( n = 1)

Lee et al, 28 2012 Case series 15 Tubal sterilization ( n = 2), salpingectomy ( n = 2), ovarian tumor enucleation( n = 1), NOTES-assisted hysterectomy ( n = 10)

Su et al, 19 2012 Case series 16 NOTES-assisted hysterectomy

Ahn et al, 14 2012, Kim, 15 2013 Case series 10 Oophorectomy ( n = 3), salpingostomy ( n = 2), salpingectomy ( n = 2), ovarian cystectomy ( n = 1), paratubal cystectomy( n = 1), ovarian wedge resection ( n = 1)

Lee et al, 13 2012 Case series 10 Tubal sterilization ( n = 3), salpingectomy ( n = 3), ovarian tumor enucleation( n = 4)

Yoshiki N. Review of transvaginal natural orifice transluminal endoscopic surgery in gynecology. Gynecol Minim Invasive Ther. 2017 Jan-Mar;6(1):1-5. doi: 10.1016/j.gmit.2016.11.007. Epub 2017 Jan 20. PMID: 30254860; PMCID: PMC6113962.

• We decided to analyze the circumference of the uterus and its weight since in our experience the uterine circumference is a major predictor for successful retrieval of the uterus through the vagina without opening and cutting of the uterus body. 

Reviewer #4: Dear Author, the article has been edited in line with the suggestions. I think the article can be published in its current form.

Reviewer #7: 

1. In the discussion, it would be appropriate to include the usual problem of morcellation with electrical instrumentation, versus morcellation with the scalpel. The electric morcellator involves mass being reduced by morcellation of tissue which is delivered to the pathologist as a meat "hamburger". The pathologist, unfortunately, is often unable to distinguish the anatomy to be analyzed and to indicate exactly the parts to be analyzed in anatomical reference of the bowel, given that he has no points of reference (PMID: 36292534). Conversely, manual morcellation with the scalpel allows the organ or mass to be removed and reconstructed with a good approximation using sutures. Therefore, I am convinced that this specification can be useful to readers of such a complex and controversial topic.

• We thank your important input; we added these points and the reference into the discussion section.

• Line 241

• " When comparing the electrical instrumentation versus morcellation with the scalpel, we can reveal one of the main limitations of the current in-bag morcellation concerning the pathology examination. The electric morcellator involves a mass being reduced, delivered to the pathologist as a slice of ground tissue. Unfortunately, the pathologist is often unable to distinguish the anatomy to be analyzed and to indicate exactly the parts to be analyzed, given that there are no reference points [28]. Conversely, manual morcellation with the scalpel allows the organ or mass to be removed and reconstructed with a good approximation using sutures. This novel tool overcomes this disadvantage providing both the anatomy preservation such as in cold knife morcellation and the efficacy of the electrical morcellation."

Reviewer #8: I read with great interest the manuscript, which falls within the aim of this Journal. In my honest opinion, the topic is interesting enough to attract the readers’ attention. Nevertheless, the authors should clarify some points and improve the discussion, as suggested below.

Authors should consider the following recommendations:

1. Manuscript should be further revised in order to correct some typos and improve style.

• We thank the reviewer for the important remark, and we corrected accordantly throughout the manuscript.

2. Authors should add further elements to discuss the role and technical differences among Intrauterine Cavity Morcellator (authors may refer to: PMID: 28948169), highlighting that these devices were not affected by the FDA warning about morcellation, which led to the adoption of endobag use as mandatory to avoid the dissemination of misdiagnosed uterine sarcomas (authors may discuss this point referring to: PMID: 35263843; PMID: 36553227).

• Thank you, we added these points into the manuscript.

• Line 267

• " Intrauterine morcellation is also familiar in the hysteroscopy field. The hysteroscopic tissue removal system seems a feasible surgical option regarding operative time and complications[34]. This hysteroscopic intrauterine morcellation was not affected by the FDA warning about morcellation in relation to the possible dissemination of misdiagnosed uterine sarcomas[35,36]. Unfortunately, the type and size of the myoma remain the biggest challenge in such a technique."

3. I recommend to highlight novel pieces of evidence about the minimally invasive management of hysterectomy for large uteri and/or for different indications, as well as the surgical specimen extraction (authors may refer to: PMID: 36498515; PMID: 34730067).

• Thank you, we added these points into the manuscript.

• Line 49- reference 4, Line 54

• " and performed with the help of the in-bag (contained) morcellation[10]."

---

## [Decision Letter · Decision Letter 2]

9 Feb 2023

An Intrauterine Cavity Morcellator: A Novel Approach to High Volume Uterus Morcellation. EX-VIVO Study

PONE-D-22-26314R2

Dear Dr. Daykan,

We’re pleased to inform you that your manuscript has been judged scientifically suitable for publication and will be formally accepted for publication once it meets all outstanding technical requirements.

Kind regards,

Andrea Giannini

Academic Editor

PLOS ONE

Additional Editor Comments (optional):

The manuscript has been modified with the comments of the reviewers. It is now ready to be published.

Reviewers' comments:

Reviewer's Responses to Questions

**Comments to the Author**

1. If the authors have adequately addressed your comments raised in a previous round of review and you feel that this manuscript is now acceptable for publication, you may indicate that here to bypass the “Comments to the Author” section, enter your conflict of interest statement in the “Confidential to Editor” section, and submit your "Accept" recommendation.

Reviewer #6: All comments have been addressed

Reviewer #8: All comments have been addressed

2. Is the manuscript technically sound, and do the data support the conclusions?

Reviewer #6: Partly

Reviewer #8: Yes

3. Has the statistical analysis been performed appropriately and rigorously? 

Reviewer #6: Yes

Reviewer #8: Yes

4. Have the authors made all data underlying the findings in their manuscript fully available?

Reviewer #6: Yes

Reviewer #8: Yes

5. Is the manuscript presented in an intelligible fashion and written in standard English?

Reviewer #6: Yes

Reviewer #8: Yes

6. Review Comments to the Author

Reviewer #6: (No Response)

Reviewer #8: I carefully evaluated the revised version of this manuscript.

Authors have performed the required changes, improving significantly the quality of the paper.

7. PLOS authors have the option to publish the peer review history of their article (what does this mean?). If published, this will include your full peer review and any attached files.

Reviewer #6: No

Reviewer #8: No

---

## [Editor Report · Acceptance letter]

7 Mar 2023

PONE-D-22-26314R2 

An Intrauterine Cavity Morcellator: A Novel Approach to High Volume Uterus Morcellation. Ex-vivo Study 

Dear Dr. Daykan:

I'm pleased to inform you that your manuscript has been deemed suitable for publication in PLOS ONE. Congratulations! Your manuscript is now with our production department. 

Kind regards, 

on behalf of

Dr. Andrea Giannini 

Academic Editor

PLOS ONE